# Topographical analysis of arterial perforators in the face: A cadaveric study

Ji–Young Son[1,2], Seung–Ho Han[1,2]*

1 Department of Anatomy, Ewha Womans University College of Medicine, Seoul, Republic of Korea,
2 Ewha Medical Academy, Ewha Womans University Medical Center, Seoul, Republic of Korea

* sanford@ewha.ac.kr

## Abstract

Arterial perforators in the face are clinically significant in plastic surgery; however, comprehensive anatomical data are limited. The aim of this study was to investigate the number, diameter, and topographic distribution of arterial perforators in the face with a diameter of 0.5 mm or greater, regarded as clinically suitable donor vessels for flap transplantation. Seven cadavers (14 hemifaces; mean age, 80.3 years) were examined using micro-computed tomography combined with dissection. The number and average diameter at the origin of each perforator were recorded, and their locations were mapped based on the Frankfort horizontal plane and vertical line through the porion. A total of 134 perforators were identified in seven major arteries: the main trunk of the facial artery, submental artery, lateral nasal branch of the facial artery, transverse facial artery, anterior auricular branch of the superficial temporal artery, zygomaticoorbital artery, and frontal branch of the superficial temporal artery. The main trunk of the facial artery exhibited the greatest number of perforators (n = 47; prevalence, 100%; average per hemiface, 3.4) and the largest mean diameter ($1.01 \pm 0.34$ mm). In the heatmap of the perforator coordinates, the frontal branch of the superficial temporal artery demonstrated the widest dispersion, with a projected area of 22.7 cm$^2$. However, the lateral nasal branch of the facial artery exhibited the highest coordinate density at 4.8 per cm$^2$. Perforators originating from the facial artery and transverse facial artery showed distinct distribution patterns relative to the horizontal line passing through the subnasale, suggesting that this line can be used as a reference in facial flap design. The comprehensive analysis of arterial perforators in the face conducted in this study provides quantitative and visualized data that may inform medical procedures of the face.

## Introduction

The arterial blood supply to the face originates from two common carotid arteries [1], and their trunks and branches form a complex vascular network.

**Data availability statement:** Access to the full raw dataset is restricted due to ethical considerations, as the data were derived from cadaveric specimens provided under the authorization of the Anatomical Dissection Review Board (ADRB) of Ewha Womans University College of Medicine. Data are available upon reasonable request and with permission from the ADRB. Requests for access should be directed to the ADRB (ADRB@ewha.ac.kr), and approval from the relevant ethics authority is required.

**Funding:** This research was supported by a grant from the Korea Technology R&D Project through the Korea Health Industry Development Institute (KHIDI), funded by the Ministry of Health and Welfare, Republic of Korea (RS-2023-KH134708). We acknowledge the support of the Ewha University-Industry Collaboration Foundation for providing English language editing assistance. The funders had no role in the study design, data collection and analysis, decision to publish, or manuscript preparation. There was no additional external funding received for this study.

**Competing interests:** The authors have declared that no competing interests exist.

The face is composed of five layers from superficial to deep: layer 1, the skin layer; layer 2, the superficial fat layer; layer 3, the musculoaponeurotic or mimic muscle layer; layer 4, the deep fat or space layer; and layer 5, the deep fascia or periosteal layer [2]. Each layer has its own distinct vasculature pattern, with the main arterial branches primarily located within layer 4 [3]. However, not all vessels remain in this layer, and some extend more superficially. Small branches, known as perforators, penetrate layer 3 perpendicularly and branch into layers 1 and 2 to supply blood to the tissues [4].

Perforators represent an anatomical consideration in reconstructive surgery to ensure sufficient tissue perfusion, particularly in flap surgeries [5]. Perforators with a diameter of at least 0.5 mm are generally considered ideal for flap application in clinical practice [6].

Successful flap transplantation critically depends on a complete understanding of the vascular anatomy [7]. Anatomical knowledge of perforators is clinically useful for selecting donor sites for reconstructive defect repair [8]. According to Yousif et al. [9], clusters of cutaneous perforators form independent and distinct skin territories that provide a structural basis for designing multiple paddle flaps for complex facial reconstruction.

Several studies have investigated the arterial perforators in the facial region. Studies focusing on perforators of the facial artery have been predominantly conducted in diverse populations [10–15], and additional reports of perforators from other arteries have also been described [16–26]. In the Korean population, specific results have been reported for the submental artery [27], transverse facial artery [28], and superficial temporal artery [29]. However, few studies have comprehensively investigated and analyzed the arterial perforators on the face.

In this study, we aimed to analyze the number, diameter, and topographic locations of arterial perforators ($\geq 0.5$ mm in diameter) in the face and to present visualized data based on these analyses.

## Materials and methods

Fourteen hemifaces from seven cadavers (six males and one female; age range, 64–101 years; mean age, $80.3 \pm 13.5$ years) were used in this study. None of the specimens showed evidence of prior surgical intervention or injury to the head or neck regions.

The 18 arterial branches examined in this study are schematically illustrated in Fig 1.

The nomenclature and classification of these vessels followed *Terminologia Anatomica Humana*, published by the International Federation of Associations of Anatomists [30].

The study protocol was approved by the Institutional Review Board (IRB) of Ewha Womans University (IRB File No. ewha-202503-0010-01), and the Anatomical Dissection Review Board (ADRB) of Ewha Womans University (ADRB File No. 25002).

Seven main steps were followed as part of the structured protocol: injection of contrast media–latex, radiographic imaging, post-processing of imaging data, dissection,

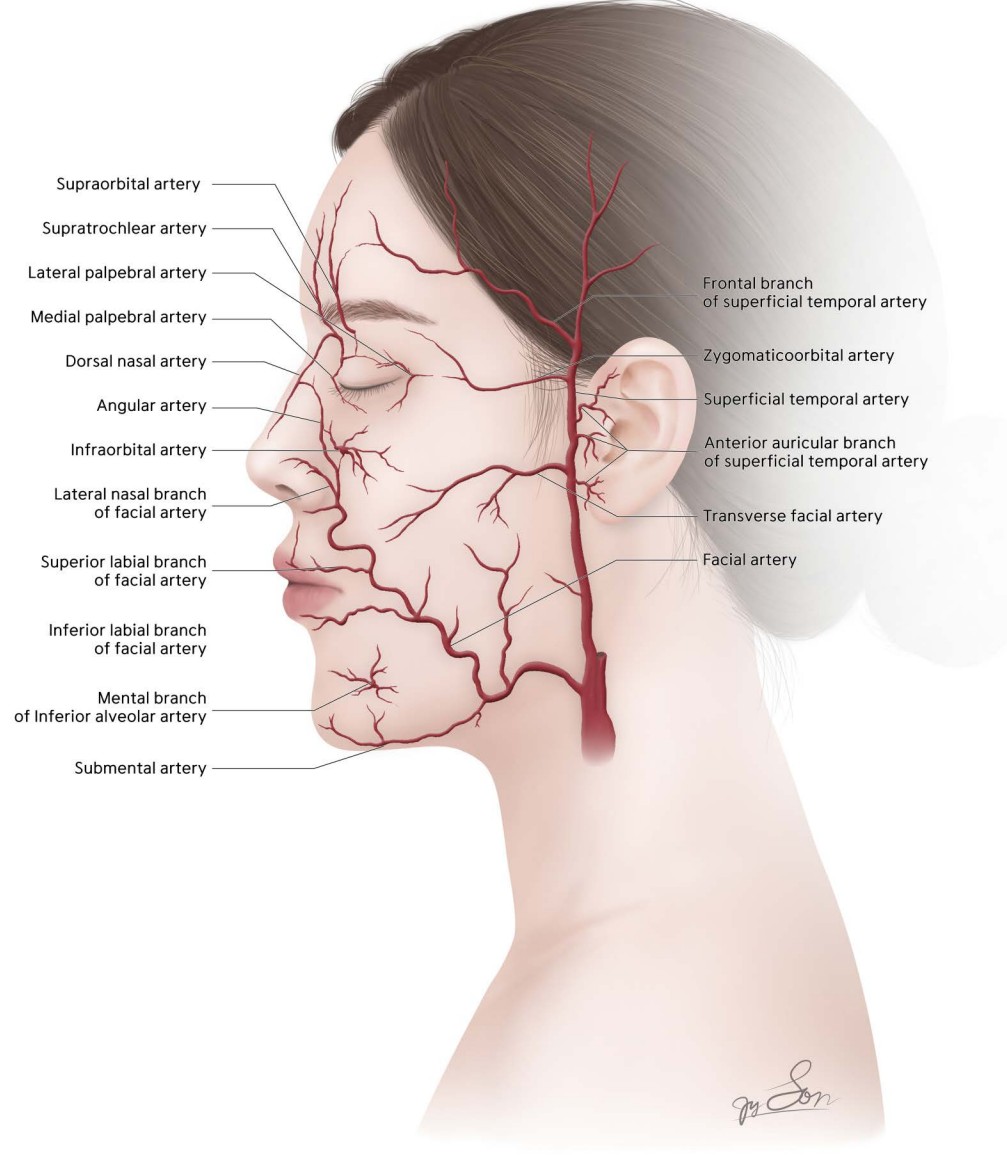

**Fig 1. Schematic representation of the arteries in the face and their branches analyzed in this study.**

identification of bony landmarks, quantitative measurements, and statistical analysis. All data were accessed for research purposes between 26/02/2025 and 30/09/2025. Image data acquisition was conducted from 26/02/2025–12/05/2025, followed by cadaveric dissection and topographical measurements from 01/03/2025–28/06/2025. A detailed description of each process is provided below.

## Contrast media–latex injection

To prepare a radiopaque injectate suitable for both radiographic imaging and dissection, a radiocontrast latex solution was formulated by mixing 10 mL of contrast medium (Angiofil®, Fumidica, Muri, Switzerland) with 40 mL of red latex (EG-LT-1010 C.V.L, Cottmo, Seoul, Republic of Korea). This 20% v/v Angiofil-latex mixture was prepared separately for

each hemiface of the soft cadavers and agitated by hand until well-blended. The common carotid artery of each specimen was cannulated and the mixture was gradually infused under continuous manual pressure. The injection was continued until noticeable resistance was felt in the syringe, and the artery was subsequently clamped to preserve the intravascular distribution of the contrast material.

## Micro–computed tomography (micro–CT) scanning

Micro-CT scanning was performed to visualize the vascular distribution in three dimensions and to obtain anatomical information on the vasculature prior to dissection. Following injection of the contrast agent, the injected heads were decapitated and positioned upright within the microtomographic device. Imaging was performed using a micro-CT scanner (Skyscan 1273®, Bruker, Kontich, Belgium) with scanning parameters detailed in Table 1.

## Post–processing

The micro-CT imaging data were reviewed and processed as three-dimensional volume-rendered images using a medical image analysis software (RadiAnt DICOM viewer; ver. 2020.2, Medixant, Poznań, Poland). A window range of –400 to +1,100 Hounsfield units was applied to optimize the visualization of contrast-filled arteries and bony structures.

## Dissection

After completion of the micro-CT scan, all hemifaces were dissected using 7.5× magnification with a surgical loupe (Looks7500, Xenosys, Incheon, Republic of Korea). Dissection was performed on all hemifaces, progressing from the skin to deeper tissues, to investigate the actual anatomical distribution of the arterial perforators. The origins of the main perforators were traced along the subcutaneous tissues and muscles.

After the initial dissection, all specimens were fixed in 10% neutral formalin solution for one week. A full-face flap, including soft tissues and periosteum was then harvested. This flap was preserved in 70% ethanol for 72 hours and re-dissected in the reverse direction, proceeding from the deep to superficial layers. The dissection results were cross-validated using micro-CT images.

### Identification of bony landmarks

To determine the distribution of major perforators, reference lines were defined by connecting fixed landmarks set in the lateral view of the anatomical position. These landmarks were designated as the orbitales and porions. The coordinates

**Table 1. Scanning parameters of the micro-CT imaging.**

| Parameter | Value |
| --- | --- |
| Image pixel size | 200.0 $\mu$m |
| Source voltage | 125 kV |
| Source current | 300 $\mu$A |
| Exposure time | 847 ms |
| Filter | Cu 1.0 mm |
| Rotation step | 0.6° |
| Object to source distance | 334.7 mm |
| Camera to source distance | 500.7 mm |
| Number of slices | 1,272 (section: 83–1,354) |
| Scan duration | 2 h 26 min 56 s |

consisted of two axes: the x-axis, the Frankfort horizontal (FH) plane connecting the orbitale and porion; and the y-axis, the vertical line passing through the FH plane (Fig 2).

In the lateral view of the face in the anatomical position, the x-axis was defined as the Frankfort horizontal plane connecting the orbitale and porion, and the y-axis as a vertical line passing through the porion. *or, Orbitale; po, Porion.*

## Measurements

Quantitative analysis was performed based on the number, diameter, and location of the perforators. These parameters were measured at the point at which the vessels emerged into the superficial fat layer after penetrating the musculoaponeurotic or mimic muscle layers. Arterial perforators with a diameter of at least 0.5 mm were documented.

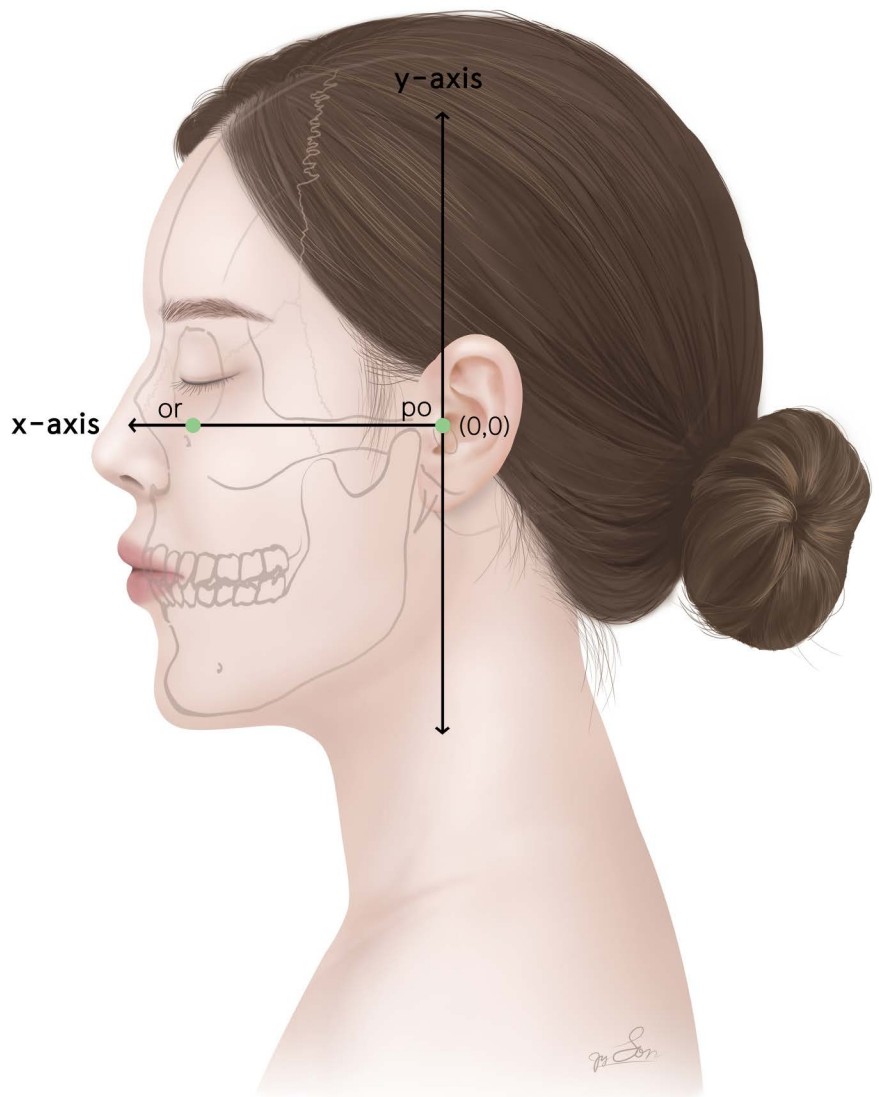

**Fig 2. Representation of the coordinate axes with bony landmarks for cutaneous projection of the arterial perforators in the face.**

### Number of arterial perforators

The number of perforators in the main arterial trunk and their prevalence per hemiface were determined using micro–CT imaging analysis and dissection.

### Diameter of arterial perforators

The outer diameters of the vessels were measured using a digital ruler within the 3D volume-rendered images. For cadaveric specimens, physical measurements were performed using digital Vernier calipers (cat. #500-181-30, Mitutoyo, Kawasaki, Japan).

### Locations of arterial perforators

The origins of the perforators were recorded as x- and y-coordinate values based on the reference lines (Fig 2), representing the surface projection. These coordinates were visualized on a topographic map.

A kernel heatmap was generated to delineate the contour lines using mapping software (Surfer; ver. 29.2.289, Golden Software, Colorado, United States). The contoured areas for each vessel were calculated using the Contour Area function of the software to identify clusters of arterial perforators.

### Statistical analysis

All statistical analyses were performed using statistical software (IBM SPSS; ver. 28.0.0.0, IBM Corp., New York, United States).

The number and diameter of the arterial perforators were calculated and are presented as mean values with standard deviations.

## Results

In this study, among the 18 arterial branches in the face (Fig 1), seven arteries with perforators measuring 0.5 mm or greater in diameter were identified: the main trunk of the facial artery (FA), submental artery (SmA), lateral nasal branch of the facial artery (LNB-FA), the transverse facial artery (TFA), anterior auricular branch of the superficial temporal artery (AAB-STA), zygomaticoorbital artery (ZoA), and frontal branch of the superficial temporal artery (FB-STA). The arterial perforators observed in a single cadaver are shown in Fig 3, with comparisons between cadaveric dissection and medical imaging.

### Number and diameter of arterial perforators

In this study, 134 perforating branches were identified across 14 hemifaces: 47 from the main trunk of the FA, nine from the SmA, 13 from the LNB-FA, 26 from the TFA, 14 from the AAB-STA, 12 from the ZoA, and 13 from the FB-STA.

Perforating branches from the main trunk of the FA exhibited a prevalence of 100%, demonstrating the highest number of branches with an average of 3.4 ± 0.9 per hemiface, and the largest mean diameter at 1.01 ± 0.34 mm (range: 0.52 to 1.79 mm). The SmA perforators were observed in eight of 14 hemifaces (prevalence: 57.1%) and a mean diameter of 0.81 ± 0.22 mm (range: 0.53 to 1.12 mm). The LNB-FA perforators showed the same prevalence (57.1%) with a slightly higher mean number (1.6 ± 0.7), but smaller mean diameter (0.72 ± 0.25 mm). The TFA perforators were present in all hemifaces (100%), with 1.9 ± 0.7 branches per hemiface and a mean diameter of 0.96 ± 0.27 mm (range: 0.54 to 1.72 mm). The AAB-STA perforators had an average of 1.3 ± 0.5 branches per side (prevalence: 78.6%), and the mean diameter was 0.68 ± 0.07 mm (range: 0.58 to 0.80 mm), representing the smallest mean diameter among the seven branches of perforators. The ZoA demonstrated an average of 1.3 ± 0.5 perforating branches (prevalence: 64.3%), with a mean diameter of 0.72 ± 0.15 mm (range: 0.51 to 1.11 mm). The FB-STA had an average of 1.2 ± 0.4 perforating branches (prevalence:

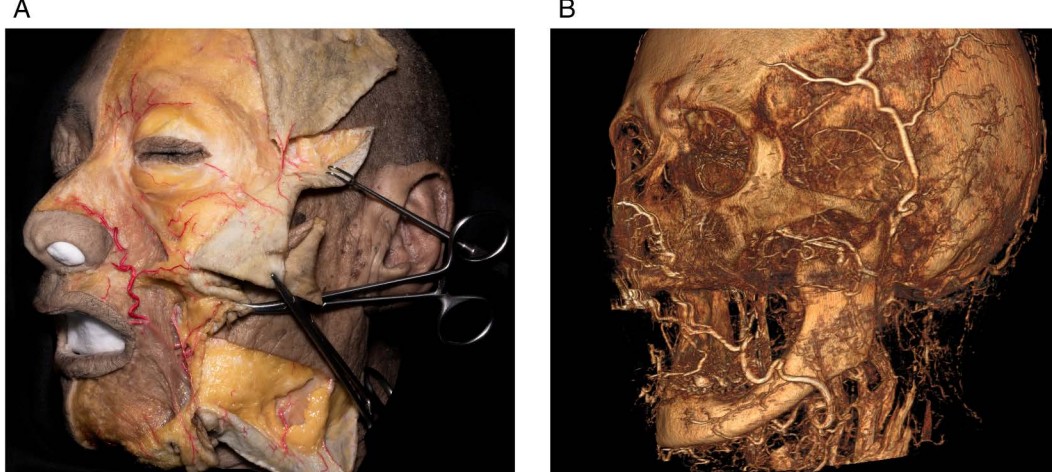

**Fig 3. Arterial perforators in a single cadaver.** (A) Photograph of arterial perforators in the face observed in the cadaver. (B) Three-dimensional rendered image of arterial perforators in the face based on medical imaging.

78.6%), with a mean diameter of 0.73 ± 0.08 mm (range: 0.63 to 0.82 mm). Data on the perforating branches of each artery are shown in Table 2.

## Locations of arterial perforators

The perforating branches were predominantly concentrated in the cheek, preauricular region, mentum, and paranasal areas. However, perforators were rarely observed in the periorbital and perioral regions. Based on the reference coordinate system, the projection plane was divided into 50 compartments (5 × 10, each measuring 20 mm × 20 mm). Among these, five compartments contained more than ten arterial perforators. Four of these high-density compartments were located in the cheek region, defined by the x-axis range of +20 to +60 mm and the y-axis range of −60 to −20 mm, where the origins of perforators from the main trunk of the FA and the TFA were predominantly distributed. The remaining compartment corresponded to the preauricular region, defined by the x-axis range of 0 to +20 mm and the y-axis range of −20–0 mm, where the origins of the AAB-STA perforators were densely clustered. The highest density was observed in the compartment defined by the x-axis range of +20 to +40 mm and the y-axis range of −60 to −40 mm, where a total of 20 perforating branches from the main trunk of the FA were observed. The overall distribution was more prominent in the inferior compartment than in the FH plane. The locations of the main perforators in each artery correspond to the anatomical course of the vessel. The distribution of the arterial perforators in the face, based on the origin of each vessel, is schematically illustrated in Fig 4.

A kernel heatmap illustrating the distribution of perforator origins is presented in Fig 5.

The projected areas within the threshold contours representing the perforator density for each vessel origin, are summarized in Table 3.

FB-STA exhibited the largest projected area (22.7 cm$^2$), followed by TFA (18.4 cm$^2$) and the main trunk of the FA (15.6 cm$^2$), all of which demonstrated widespread distributions. In contrast, SmA and LNB-FA showed relatively localized distributions, with projected areas of 2.0 cm$^2$ and 2.7 cm$^2$, respectively. Although the main trunk of the FA had the largest number of perforator origins (47 coordinates), its projected area was smaller than that of the TFA. Conversely, FB-STA exhibited the lowest density value (0.6 per cm$^2$), indicating a highly dispersed pattern. In contrast, SmA,

**Table 2. Number, prevalence, and diameter of arterial perforators in the face.**

| Artery | Prevalence (%)[a] | Mean number of perforators ± SD | Range of number | Mean diameter ± SD (mm) |
|---|---|---|---|---|
| **FA** | | | | |
| Main trunk of the FA | 100.0 | 3.4 ± 0.9 | 2–5 | 1.01 ± 0.34 |
| SmA | 57.1 | 1.1 ± 0.4 | 1–2 | 0.81 ± 0.22 |
| LNB-FA | 57.1 | 1.6 ± 0.7 | 1–3 | 0.72 ± 0.25 |
| **STA** | | | | |
| TFA | 100.0 | 1.9 ± 0.7 | 1–3 | 0.96 ± 0.27 |
| AAB-STA | 78.6 | 1.3 ± 0.5 | 1–2 | 0.68 ± 0.07 |
| ZoA | 64.3 | 1.3 ± 0.5 | 1–2 | 0.72 ± 0.15 |
| FB-STA | 78.6 | 1.2 ± 0.4 | 1–3 | 0.73 ± 0.08 |

[a]Prevalence (%) represents the percentage of hemifaces (n = 14) in which arterial perforators measuring ≥0.5 mm were identified from each vessel, and indicates how many of the 14 hemifaces had at least one perforator from the specified vessel. This value was calculated as follows: (number of hemifaces with arterial perforators/14) × 100.

*AAB-STA, Anterior auricular branch of the superficial temporal artery; FA, Facial artery; FB-STA, Frontal branch of the superficial temporal artery; LNB-FA, Lateral nasal branch of the facial artery; SD, Standard deviation; SmA, Submental artery; STA, Superficial temporal artery; TFA, Transverse facial artery; ZoA, Zygomaticoorbital artery.*

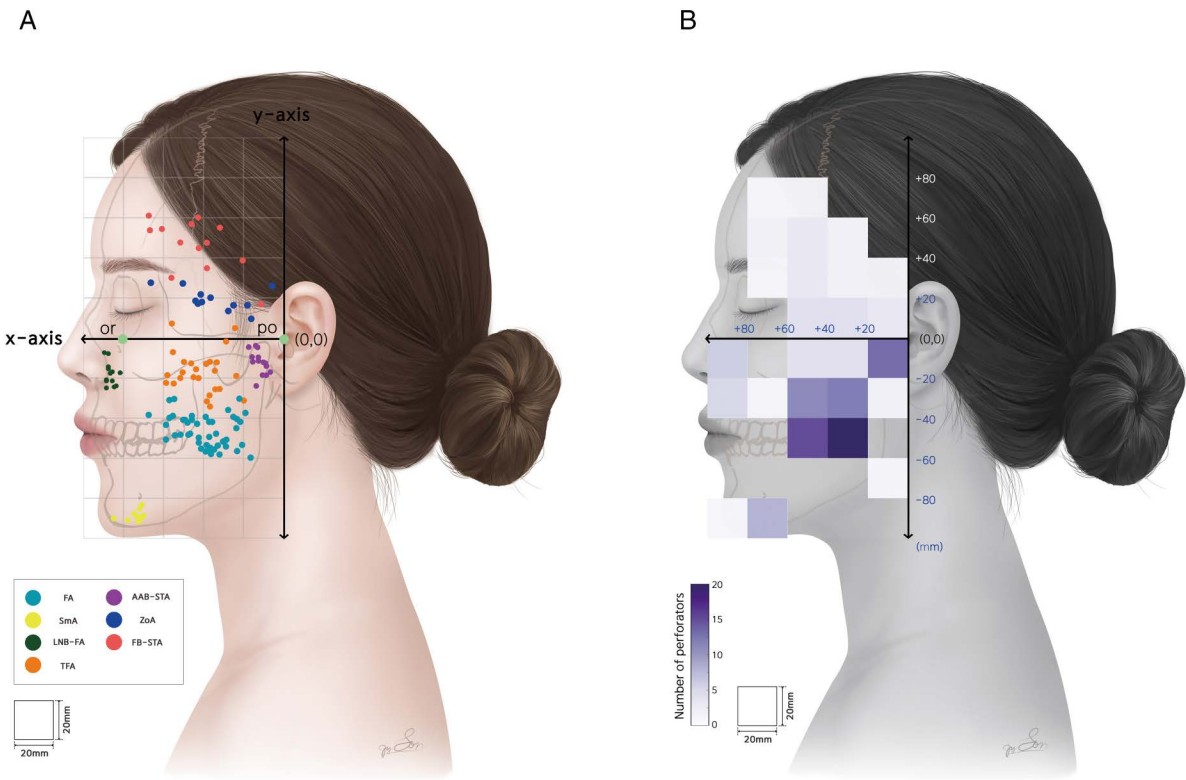

**Fig 4. Topographic representation of the perforator origin distribution.** (A) Each colored dot represents the main perforator origin from a different artery: cyan, main trunk of the FA; yellow, SmA; dark green, LNB-FA; orange, TFA; violet, AAB-STA; blue, ZoA; crimson, FB-STA. (B) Density map constructed based on the number of coordinate points in each region, overlaid onto the 50-sector topographic grid. Darker shades represent higher densities of the perforator origins. *AAB-STA, Anterior auricular branch of the superficial temporal artery; FA, Facial artery; FB-STA, Frontal branch of the superficial temporal artery; LNB-FA, Lateral nasal branch of the facial artery; or, Orbitale; po, Porion; SmA, Submental artery; TFA, Transverse facial artery; ZoA, Zygomaticoorbital artery.*

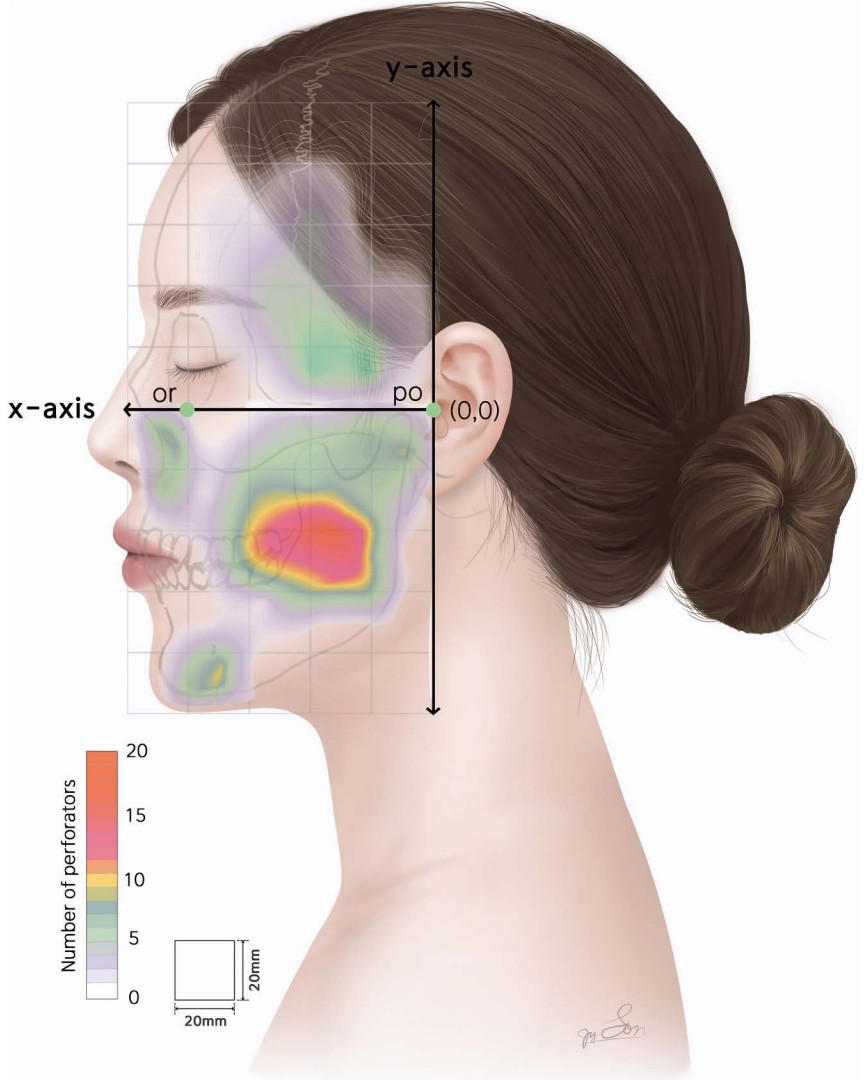

**Fig 5. Kernel density map of the distribution of the perforators.** Warmer colors (red to magenta) indicate higher densities of perforator origins, while cooler colors (light purple to white) indicate lower densities. *or, Orbitale; po, Porion.*

LNB-FA, and AAB-STA showed densities ≥ 4.0 per cm$^2$, suggesting a concentrated distribution of arterial perforators within narrow areas (Table 4).

## Discussion

Although the anatomy of the arterial system in the face has been well-studied, its clinical application remains limited and involves uncertainties. Textbooks and literature in the field of anatomy typically present average vascular distributions, often overlooking individual variations and courses of perforators [31]. For example, illustrations in a textbook [32] simplify the main arteries, potentially causing confusion among clinicians when unexpected vasculature is encountered during medical procedures. According to Pilsl et al. [33], the course and distribution of the facial artery correspond to descriptions in textbooks in approximately 40% of cases.

**Table 3. Projected area, number of perforator origins, and density of arterial perforators in the face.**

| Artery | Projected area (cm²)[a] | Total number of perforators (n)[b] | Density (n/cm²) |
|---|---|---|---|
| **FA** | | | |
| Main trunk of the FA | 15.6 | 47 | 3.0 |
| SmA | 2.0 | 9 | 4.5 |
| LNB-FA | 2.7 | 13 | 4.8 |
| **STA** | | | |
| TFA | 18.4 | 26 | 1.4 |
| AAB-STA | 3.3 | 14 | 4.2 |
| ZoA | 14.8 | 12 | 0.8 |
| FB-STA | 22.7 | 13 | 0.6 |

a The projected area (cm²) refers to the flat surface area of clustered arterial perforators in the face based on kernel heatmap analysis.

b Total number of perforators (n) corresponds to perforator origins (diameter: ≥ 0.5 mm) identified for each artery.

*AAB-STA, Anterior auricular branch of the superficial temporal artery; FA, Facial artery; FB-STA, Frontal branch of the superficial temporal artery; LNB-FA, Lateral nasal branch of the facial artery; n, Total number of arterial perforators; SmA, Submental artery; STA, Superficial temporal artery; TFA, Transverse facial artery; ZoA, Zygomaticoorbital artery.*

**Table 4. Relative dispersion pattern of arterial perforators in the face based on density (n/cm²).**

| Dispersion[a] | Range of density (n/cm²) | Arteries included |
|---|---|---|
| High dispersion (low density) | <1 | ZoA, FB-STA |
| Intermediate | 1–4 | Main trunk of the FA, TFA |
| Low dispersion (high density) | >4 | SmA, LNB-FA, AAB-STA |

[a]Degree of dispersion was classified into three levels: high dispersion, intermediate, and low dispersion.

*AAB-STA, Anterior auricular branch of the superficial temporal artery; FA, Facial artery; FB-STA, Frontal branch of the superficial temporal artery; LNB-FA, Lateral nasal branch of the facial artery; n, Total number of arterial perforators; SmA, Submental artery; TFA, Transverse facial artery; ZoA, Zygomaticoorbital artery.*

There have been a total of five published books focusing on the arterial perforators of the face [34–38]. Perforating branches of the face remain a subject of ongoing interest for researchers. Therefore, several individual studies have examined these branches. A summary of these studies is presented in Table 5. There are several discrepancies in this study compared to the findings reported in the literature.

First, the number of arterial perforators from the main trunk of the FA was lower than that reported previously. However, in this study, the LNB-FA was categorized separately, and the perforators originating from this branch were counted independently. When considered together, the overall count was comparable to previously reported values. This result may reflect the more detailed anatomical subdivision applied in this study to branches originating from the FA.

Second, the prevalence of specific perforators, such as SmA, LNB-FA, AAB-STA, ZoA, and FB-STA, was lower than that in earlier reports. These discrepancies may be attributed to the measurement criteria used in this study. Only perforators with a diameter of 0.5 mm or greater, which are clinically relevant, were included in the analysis, and smaller branches have been excluded. As a result, branches smaller than 0.5 mm in diameter with limited clinical utility might have been excluded. The criterion of 0.5 mm as the minimum diameter for clinically relevant perforators is well established in the literature [39–43], although higher thresholds have been proposed. For instance,

**Table 5. Comparison between previous studies and the present study on arterial perforators in the face.**

| Artery | Author(year) | Sample source | Sample size: hemifaces | Population | Prevalence (%) | No. of APs | Filter criteria: diameter (mm) | Mean diameter of APs (mm) |
|---|---|---|---|---|---|---|---|---|
| FA | This study(2025) | Soft cadavers | 14 | Korean | 100.0 | Main trunks of the FA: 47(total), 3.4(per hemiface) | ≥0.5 | 1.01 |
| | Hofer et al.(2005) [10] | Soft cadavers | 10 | Dutch | 100.0 | 5.7(per hemiface) | – | 1.20 |
| | Ng et al.(2010) [11] | Fixed cadavers | 16 | British | 100.0 | 4(per hemiface) | – | 0.94 |
| | Qassemyar et al.(2012) [12] | Soft cadavers | 20 | French | – | 5.05(per hemiface) | >0.5 | 0.96 |
| | deTaddéo et al.(2014) [13] | Soft cadavers | 20 | French | – | 5.65(per hemiface) | ≥0.5 | 0.80 |
| | Camuzard et al.(2015) [14] | Soft cadavers | 20 | French | – | 6(per hemiface) | >0.5 | 0.91 |
| | Kandathil et al.(2024) [15] | Soft cadavers | 20 | Austrian | 100.0 | 4.45(per hemiface) | ≥0.5 | 0.65 |
| SmA | This study(2025) | Soft cadavers | 14 | Korean | 57.1 | 9(total), 1.1(per hemiface) | ≥0.5 | 0.81 |
| | Kim et al.(2002) [27] | Patients / Soft cadavers | 16 / 8 | Korean / Japanese | 75.0 | 1-2(per hemiface) | – | – |
| | Atamaz Pinar et al. (2005) [16] | Fixed cadavers | 50 | Turkish | 60.0 | 30(total) | – | 1.40 |
| | Tang et al.(2011) [17] | Soft cadavers | 20 | Chinese | 100.0 | 1.8(per hemiface) | – | ≥0.5 |
| LNB-FA | This study(2025) | Soft cadavers | 14 | Korean | 57.1 | 13(total), 1.6(per hemiface) | ≥0.5 | 0.72 |
| | Lombardo et al.(2016) [18] | Soft cadavers | 16 | Italian | – | – | – | 0.91 |
| | Posso et al.(2018) [19] | Soft cadavers | 9 | Colombian | 100.0 | 9(total), 2.3(per hemiface) | – | 1.00–1.50 |
| STA | This study(2025) | Soft cadavers | 14 | Korean | 78.6 | AAB-STA: 14(total), 1.3(per hemiface) | ≥0.5 | 0.68 |
| | | | | | 78.6 | FB-STA: 13(total), 1.2(per hemiface) | | 0.73 |
| | Jo et al.(2012) [29] | Patients / Soft cadavers | 18 / 10 | Korean | 100.0 | 18 / 10(total) | – | 0.70 |
| | Xu et al.(2014) [20] | Fixed cadavers | 20 | Chinese | 85.0 | AAB-STA: 17(total) | – | 0.65 |
| | Aveta et al.(2017) [21] | Patients / Soft cadavers | 100 / 14 | Italian | 100.0 | FB-STA: 2.5(per hemiface) | – | – |
| TFA | This study(2025) | Soft cadavers | 14 | Korean | 100.0 | 26(total), 1.9(per hemiface) | ≥0.5 | 0.96 |
| | Bozikov et al.(2008) [22] | Fixed cadavers | 24 | British | 100.0 | 24(total) | – | 0.65 |
| | Yang et al.(2010) [28] | Fixed cadavers | 44 | Korean | 100.0 | 2.0(per hemiface) | – | – |
| | Pierrefeu et al.(2019) [23] | Soft cadavers | 14 | French | 85.7 | 23(total), 1.64(per hemiface) | – | 1.01 |
| | Zhang et al.(2023) [24] | Patients | 18 | Chinese | 100.0 | At least 1(per hemiface) | – | – |
| ZoA | This study(2025) | Soft cadavers | 14 | Korean | 64.3 | 12(total), 1.3(per hemiface) | ≥0.5 | 0.72 |
| | Bozikov et al.(2008) [22] | Fixed cadavers | 24 | British | 79.2 | 19(total) | – | 0.40 |
| | Chen et al.(2023) [25] | Soft cadavers | 87 | Chinese | 86.1 | 71(total) | – | – |

*(Continued)*

**Table 5.** (Continued)

| Artery | Author(year) | Sample source | Sample size: hemifaces | Popula-tion | Preva-lence (%) | No. of APs | Filter criteria: diameter (mm) | Mean diameter of APs (mm) |
|---|---|---|---|---|---|---|---|---|
| IoA | Hufschmidt et al. (2019) [26] | Soft cadavers | 23 | French | 100.0 | Mucosal perforators: 3.16(per hemiface) | ≥0.4 | 0.74 |

APs, Arterial perforators; AAB-STA, Anterior auricular branch of the superficial temporal artery; FA, Facial artery; FB-STA, Frontal branch of the superficial temporal artery; IoA, Infraorbital artery; LNB-FA, Lateral nasal branch of the facial artery; SmA, Submental artery; STA, Superficial temporal artery; TFA, Transverse facial artery; ZoA, Zygomaticoorbital artery.

Seo et al. [44] recommended 0.7 mm for the inguinal and perineal regions; Gravvanis et al. [45] and Drimouras et al. [46] suggested 1.0 mm for the abdominal and lower limb regions, respectively; Yoon et al. [47] proposed 1.0 mm for the gluteal region; and Rozen et al. [48] identified deep inferior epigastric artery perforators larger than 1.5 mm as major vessels suitable for flap reconstruction. However, these studies examined anatomical regions larger than the facial region. Given that this study focused on the facial area, the smallest threshold was selected to enable a more detailed anatomical observation. Another factor is that the contrast media–latex mixture was manually infused, resulting in potential variability in perfusion pressure among the specimens. This variation could have limited the filling of the distal arterial branches because manual injection does not ensure constant intraluminal pressure. Furthermore, pre-existing intravascular blood clots may have impeded the flow of contrast agents or latex, thereby reducing the visibility of smaller perforators. Similarly, previous cadaveric studies noted that inadequate perfusion can hinder the delivery of the injectate to terminal vessels, thereby influencing the number of identifiable arteries [49,50].

Third, among previous studies, Hufschmidt et al. [26] examined the perforators of the infraorbital artery (IoA), and their analysis focused on mucosal perforators rather than cutaneous ones. Mucosal branches arising from the IoA typically penetrate deeper facial layers; however, this study focused on perforators oriented toward the superficial cutaneous plane.

Fourth, among the 20 previous studies [10–29] reviewed, five studies (Jo et al. [29], Qassemyar et al. [12], Kandathil et al. [15], Pierrefeu et al. [23], Chen et al. [25]) provided information on the location and territory of the arterial perforators. In this study, a novel mapping method was applied to analyze the distribution of perforators based on their anatomical locations, which were visualized accordingly.

The distribution patterns of the perforators from the main trunks of the FA and TFA were well distinguished by a horizontal line passing through the subnasale parallel to the FH plane. This horizontal line provided an anatomical boundary with the TFA perforator region located above and the FA perforators below (Fig 6).

Most main perforators of the TFA (88.5%) were above the horizontal line passing through the subnasale. Only a few (11.5%) were located below this line. Although both arteries originate from the common carotid artery, the external carotid artery bifurcates into the FA and superficial temporal artery, thereby establishing distinct vascular territories with clinically meaningful differences in perforator distribution.

A comprehensive analysis of the arterial perforator distribution in the face was conducted using consistent criteria. While previous studies have largely focused on describing perforators from single vessels or within specific anatomical regions, this study quantitatively presents an overview of the topographical distribution across multiple vascular groups, including the main trunks of the FA, SmA, LNB-FA, TFA, AAB-STA, ZoA, and FB-STA. The distribution of these perforators can provide anatomical guidance for flap design before clinical procedures. The findings of this study, indicating that perforators are predominantly clustered in the cheek, mentum, preauricular and paranasal areas, are consistent with the regions where flap design is frequently performed in clinical practice. From the donor perspective, these high-density regions provide a favorable anatomical basis for the design of local flaps, such as the nasolabial, periauricular, and submental flaps. The use of the perforator map enables selective flap harvest, which may contribute to reduced intraoperative

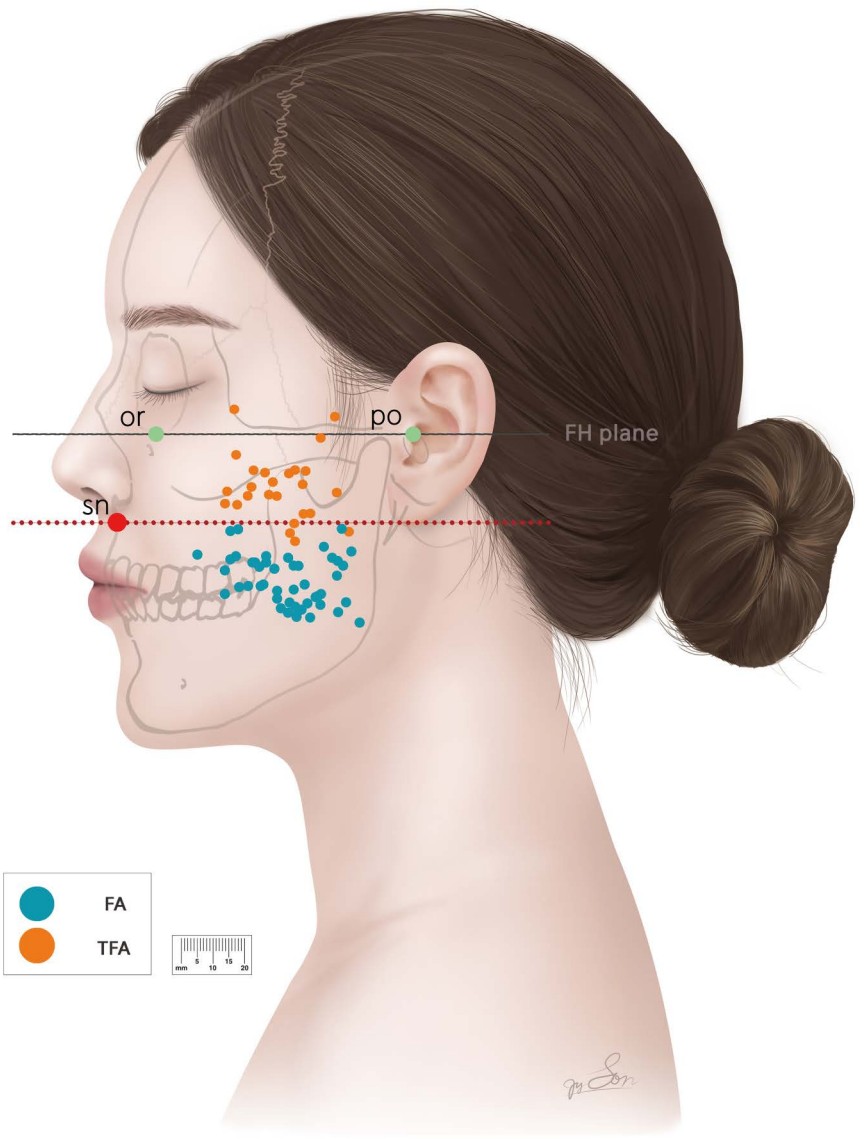

**Fig 6. Distribution of perforating branches from the TFA and the main trunks of the FA in relation to the subnasale.** Each colored dot represents the main perforator from a different artery: cyan, main trunk of the FA; orange, TFA. A horizontal line (red dotted line) passes through the subnasale parallel to the FH plane. *FA, Facial artery; FH plane, Frankfort horizontal plane; or, Orbitale; po, Porion; sn, Subnasale; TFA, Transverse facial artery.*

bleeding, decreased donor-site morbidity, and minimization of unnecessary skin incisions over vascular territories. From the recipient perspective, the perfusion territory that can be achieved after flap transfer can be estimated based on the distribution of perforators.

One of the limitations of this study is the lack of quantitative analysis of the perforasomes, the cutaneous territories supplied by the perforators of the face. As perforators emerge from their sources and ascend toward more superficial layers, their diameters progressively decrease, forming a dense vascular network. Although arterial perforators exhibit a tree-like distribution throughout the skin, further experimental methodologies such as ink injection and thermography, as previously described by Whetzel et al. [51] and Chijiwa et al. [52], are required to accurately delineate their vascular territories.

Therefore, additional research is necessary to apply these findings in clinical settings. Subsequent research should specifically focus on the precise quantification of the perforasome territories. Thorough analyses, using various advanced imaging modalities and experimental approaches, are expected to provide a fundamental basis for clinical procedures.

## Conclusion

We conducted a detailed topographical analysis of arterial perforators in the face. By constructing a topographical map based on the anatomical data of the arterial perforators in the face, this study provides fundamental data that can be clinically utilized.

The present analysis offers a broader view through a quantitative evaluation of the entire face. Accuracy was improved by integrating micro-CT imaging with dissection, and the clinical relevance was enhanced by focusing on perforators measuring 0.5 mm or greater, which are considered clinically significant.

These findings support flap design and minimize vascular damage during facial reconstruction. Quantitative localization of clinically relevant arterial perforators may help surgeons navigate vascular maps and improve procedural safety during facial interventions.

## Acknowledgments

We would like to express our sincere gratitude to the individuals who donated their bodies to medical science.

The authors acknowledge the Ewha University–Industry Collaboration Foundation for assistance with English language editing.

## Author contributions

**Conceptualization:** Seung-Ho Han.

**Data curation:** Ji–Young Son.

**Formal analysis:** Ji–Young Son.

**Funding acquisition:** Seung-Ho Han.

**Investigation:** Ji–Young Son, Seung-Ho Han.

**Methodology:** Ji–Young Son.

**Project administration:** Seung-Ho Han.

**Resources:** Seung-Ho Han.

**Supervision:** Seung-Ho Han.

**Validation:** Ji–Young Son, Seung-Ho Han.

**Visualization:** Ji–Young Son.

**Writing – original draft:** Ji–Young Son.

**Writing – review & editing:** Ji–Young Son, Seung-Ho Han.

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
