## [Decision Letter · Decision Letter 0]

10 Mar 2026

PONE-D-25-68981Topographical Analysis of Arterial Perforators in the Face: A Cadaveric StudyPLOS One

Dear Dr. Han,

Thank you for submitting your manuscript to PLOS ONE. After careful consideration, we feel that it has merit but does not fully meet PLOS ONE’s publication criteria as it currently stands. Therefore, we invite you to submit a revised version of the manuscript that addresses the points raised during the review process.

Dear Authors,

Thank you for submitting your manuscript entitled “Topographical Analysis of Arterial Perforators in the Face: A Cadaveric Study” to PLOS ONE.

First, I would like to apologize for the delay in reaching a decision on your manuscript. Unfortunately, it proved difficult to secure reviewers within the expected timeframe, and I appreciate your patience during the review process.

The reviewer considered the manuscript to be technically sound and well written, and the data appear to support the conclusions presented. The study provides useful anatomical information regarding the distribution of facial arterial perforators.

However, before the manuscript can be accepted for publication, a small number of minor revisions should be addressed. In particular, the reviewer suggests expanding the Discussion to better relate the anatomical findings to commonly used facial flaps (for example nasolabial, submental, or lateral orbital flaps). In addition, please review the manuscript for minor language corrections and consider slightly reducing redundancy in parts of the Discussion.

Therefore, my decision is **Minor Revision**.

Please submit a revised manuscript together with a point-by-point response to the reviewer’s comments indicating how each point has been addressed.

I look forward to receiving your revised manuscript.

We look forward to receiving your revised manuscript.

Kind regards,

Luis A Arráez-Aybar

Academic Editor

PLOS One

Journal Requirements:

“This research was supported by a grant of the Korea Technology R&D Project through the Korea Health Industry Development Institute (KHIDI), funded by the Ministry of Health & Welfare, Republic of Korea (RS-2023-KH134708).”

“This research was supported by a grant of the Korea Technology R&D Project through the Korea Health Industry Development Institute (KHIDI), funded by the Ministry of Health & Welfare, Republic of Korea (RS-2023-KH134708).”

4. We note that your Data Availability Statement is currently as follows: [All relevant data are within the manuscript.]

Additional Editor Comments (if provided):

Thank you for submitting your manuscript entitled “Topographical Analysis of Arterial Perforators in the Face: A Cadaveric Study.” The reviewer considered the manuscript to be technically sound and well written, and the data appear to support the conclusions presented. The study provides a useful anatomical overview of facial arterial perforators using a combination of micro-CT imaging and cadaveric dissection.

Before the manuscript can be accepted, please address the following minor points raised during review and editorial assessment.

1. Relationship with clinically used facial flaps

The reviewer suggests that the Discussion would benefit from a clearer connection between the anatomical findings and commonly used facial flaps. Please consider briefly expanding the Discussion to relate your results to well-known flap techniques such as the nasolabial flap, submental flap, or lateral orbital/periorbital flaps. Clarifying how the mapped perforator distributions may support the design or safety of these flaps would improve the clinical contextualization of the study.

2. Clarity and concision of the Discussion

Some sections of the Discussion repeat information already presented in the Results or the literature review. Please consider slightly reducing redundancy and focusing the Discussion on the interpretation and implications of the findings.

3. Minor language and typographical corrections

The manuscript is generally well written. However, please review the text carefully to correct minor grammatical or typographical inconsistencies and ensure consistency in abbreviations and formatting throughout the manuscript.

Please submit a revised version of the manuscript together with a detailed point-by-point response explaining how each comment has been addressed.

Reviewers' comments:

Reviewer's Responses to Questions

**Comments to the Author**

1. Is the manuscript technically sound, and do the data support the conclusions?

Reviewer #1: Yes

2. Has the statistical analysis been performed appropriately and rigorously?

Reviewer #1: N/A

3. Have the authors made all data underlying the findings in their manuscript fully available?

Reviewer #1: Yes

4. Is the manuscript presented in an intelligible fashion and written in standard English?

Reviewer #1: Yes

5. Review Comments to the Author

Reviewer #1: Well written paper.

It would be beneficial to include discussions that specifically address the relationship between well-known flaps(e.g. nasolabial fold flap, submental flap, or lateral orbital flap) and perforators.

6. PLOS authors have the option to publish the peer review history of their article (what does this mean?). If published, this will include your full peer review and any attached files.

Reviewer #1: **Yes:** Shimpei Miyamoto

---

## [Author Response · Author response to Decision Letter 1]

16 Apr 2026

We sincerely thank the editor and reviewer for the careful review of our manuscript and for the constructive comments that helped us improve the quality and clarity of the manuscript. We have revised the manuscript accordingly, and our detailed responses are provided below.

Comment 1. Relationship with clinically used facial flaps

Response 1.:

Thank you for this valuable comment. In response, we have expanded the Discussion section to clearly describe the clinical relevance of our findings in relation to commonly used facial flaps. Specifically, we added a new paragraph (Lines 243–253) explaining how the perforator distribution map can be applied in clinical practice.

In this revision, we emphasized that the identified high-density perforator regions correspond to areas frequently used in flap design, and we described their implications from both donor and recipient perspectives. These additions aim to provide a more concrete explanation of how the anatomical findings can be translated into practical surgical applications.

Comment 2. Clarity and concision of the Discussion

Response 2.:

We appreciate the editor’s suggestion to improve the clarity and conciseness of the Discussion section. In response, we have carefully revised the section by removing repetitive descriptions of numerical results that were already presented in the Results section.

In addition, references related to clinically meaningful perforator diameter thresholds have been reorganized and presented in a more concise manner to improve readability and allow for easier comparison.

Comment 3. Minor language and typographical corrections

Response 3.:

Following the editor’s suggestion, the manuscript has undergone professional English language editing. We have corrected grammatical errors and typographical inconsistencies throughout the manuscript to improve overall clarity and readability.

We hope that these revisions have adequately addressed your comments and improved the manuscript. We sincerely appreciate your consideration.

---

## [Editor Report · Decision Letter 1]

23 Apr 2026

Topographical analysis of arterial perforators in the face: A cadaveric study

PONE-D-25-68981R1

Dear Dr. Han,

We’re pleased to inform you that your manuscript has been judged scientifically suitable for publication and will be formally accepted for publication once it meets all outstanding technical requirements.

Kind regards,

Luis A Arráez-Aybar

Academic Editor

PLOS One

Additional Editor Comments (optional):

Dear Authors

I am pleased to inform you that your manuscript entitled "Topographical Analysis of Arterial Perforators in the Face: A Cadaveric Study" has been accepted for publication in PLOS ONE.

The revised version satisfactorily addresses the reviewers’ and editorial comments and meets the journal’s criteria for publication.

The manuscript will now proceed to production, and you will be contacted with further instructions.
---

## [Editor Report · Acceptance letter]

PONE-D-25-68981R1

PLOS One

Dear Dr. Han,

I'm pleased to inform you that your manuscript has been deemed suitable for publication in PLOS One. Congratulations! Your manuscript is now being handed over to our production team.

Kind regards,

on behalf of

Dr. Luis A Arráez-Aybar

Academic Editor

PLOS One